# Teachers' Global Perceptions and Views, Practices and Needs in Multicultural Settings

**Zoe Karanikola \*, Glykeria Katsiouli and Nektaria Palaiologou**

Language Education for Refugees and Migrants (L.R.M.) Programme, Hellenic Open University, Perivola Patron, 26335 Patras, Greece; std506669@ac.eap.gr (G.K.); nekpalaiologou@eap.gr (N.P.)
\* Correspondence: karanikola.zoi@ac.eap.gr; Tel.: +30-261-096-2815

**Abstract:** Multiculturalism and globalization are common traits of western societies, and affect the way people interact and communicate. In such a context, this study comes to investigate teachers' perceptions, practices and needs towards global and intercultural competences. This study was designed and implemented in order to shed light on major issues which are associated with the context of global competences as an umbrella term, which arose during the researchers' participation at an Erasmus plus European project. The research took place at the region of Attica and thirteen teachers (N = 13) of reception classes—Zones of Educational Priority (ZEP)—participated in the interview. A qualitative case study followed, focusing on a specific geographic region, and the semi-structured interview tool was used. The findings of the research indicate that educators do not feel certain about the differences between global and intercultural competences. However, they consider that these competences are of great importance and they recognize the contribution of schools to their development. Regarding their practices, they mostly refer to the dialogue and discussion techniques, the role playing, and the project-based teaching. Finally, participants consider that they are not sufficiently prepared to teach global and intercultural competences. Thus, participation in relevant training programs is important.

**Keywords:** global competences; intercultural competences; educational priority zones; refugee and immigrant students; teaching practices; teachers' needs

## 1. Introduction

Europe is facing an unprecedented migrant crisis, since hundreds of thousands of refugees and migrants continue to make their way across the Mediterranean to Europe. In addition, the world looks like a global village, as it is estimated that more than 250,000,000 people on the planet live in different countries from those in which they were born [1]. This primarily happens due to the great developmental differences between the countries in terms of their economic and social development, the rapid growth of the world population since the beginning of the 19th century, the great changes that the world map was subject to as a consequence of the Second World War, the rapid technological developments, the advent of the 4th industrial revolution and globalization [2].

Globalization makes people act and interact with others in global, diverse and complex environments [3], which bring about new needs, demands and conditions. Thus, global and intercultural competences are considered to be of major importance, since they help individuals respect each other and cooperate efficiently [4].

Education should go hand in hand with the social and working demands, as one of its purposes is to prepare individuals to recognize and get adjusted to the global socioeconomic and environmental changes [3]. Cummins [5] argues that schools should give the chance and the motive to teachers to learn how to teach culturally and linguistically diverse students in an appropriate way. What is more, some important factors that could indicate the level of students' acquisition of global or intercultural competence are teachers'

perceptions and views on diversity, their appropriate knowledge and relevant skills [6]. Skills that promote cultural awareness, acceptance, adaptability and self-development, targeting the general well-being and sustainable development seem to be necessary for the preparation of future global citizens [4].

The preparation of global and intercultural competent citizens is an issue that concerns Greece, as it is a country that receives a notable number of migrants and refugees [7]. According to the Operational Data Portal [8], which was created in 2011 in order to provide information and data regarding refugees' flows and facilitate coordination of their emergencies, 74,613 refugees arrived in our country in 2019 and 15,696 arrived in 2020. The most common countries of origin are Afghanistan, Somalia, the State of Palestine, Iraq and Syria.

In addition, by mainly studying the Greek legislation for the period 1996–2020, we find that there is a plethora of laws, presidential decrees and ministerial decisions regarding the education of immigrants and refugees. Legal texts emphasize the education of people belonging to different background and the promotion of human rights. The 1996 law (2413/96) was the first institutional measure taken towards the direction of intercultural education, and it expressed the intention of the Greek policy to set aside assimilation approaches and to cope with the linguistic and cultural needs of all students. This legislation framework introduced the term of "intercultural education" for the first time here in Greece, established twenty six intercultural schools and affected the development of related in service and pre-service training programs and the writing of new curricula and textbooks [9].

However, despite the fact that teachers have both knowledge and willingness to teach in this direction, in many cases they do not use this knowledge due to inadequate environmental support [10]. Finally, teachers are members of the dominant culture, so they may face difficulties in understanding the challenges, fears and needs of individuals who are called to get adjusted to culturally diverse environments [11].

This study was designed and implemented in order to shed light on major issues which are associated with teachers' global and intercultural competences as an umbrella term, focusing on teachers who work with refugee and immigrant students. These issues arose during the researchers' participation at an Horizon project.

## 2. Global and Intercultural Competence

Global competence seems to be the foundation of employability and citizenship in a global era [4], and it is included among the new abilities that members of the workforce need in order to function successfully in a world of growing diversity and complexity [12]. As a term it is widely used among academics and scholars. However, there is a strong disagreement on what exactly constitutes global competence [13,14]. Bennett's developmental model of intercultural sensitivity is often credited as the theoretical foundation of the development of global competence [15]. Specifically, intercultural sensitivity is one's ability to discern and experience relevant cultural differences and is a key prerequisite for both intercultural and global competence [9].

Hunter [16] (p. 1) tries to define global competence as having an open mind while actively seeking to understand the cultural norms and expectations of others and leveraging this gained knowledge to interact, communicate and work effectively in diverse environments. In addition, the OECD [17] applies the multidimensional concept of global competence in terms of maximizing economic gains, care for the environment, social harmony, and establishing acceptable levels of security, health, and education. Specifically, it defines global competence as the capacity to examine local, global, and intercultural issues; to understand and appreciate the perspectives and world views of others; to engage in open, appropriate, and effective interactions with people from different cultures; and to act for collective well-being and sustainable development [4]. In addition, it provides us with four key aspects of global competence. The first is to investigate the world beyond one's immediate environment by examining issues of local, global, and cultural significance. The second is to recognize, understand, and appreciate the perspectives and world views of

others. The third is to communicate ideas effectively with diverse audiences by engaging in open, appropriate and effective interactions across cultures, and the fourth is to take action for collective well-being and sustainable development both locally and globally.

Currently, there are many countries that attend to integrate global competence in their educational systems [4]. With the rise of technology, global communications and interconnectedness of the world have extended, making students' development of global competence an urgent need. Moreover, global competence is related to individuals' effort to live cooperatively and to be employed in multicultural contexts. Towards this direction, in 2018, the member states of both the United Nations, through its adoption of the Sustainable Development Goals, and the OECD, through its Programme for International Student Assessment (PISA), prioritized education for global citizenship and global competence [4].

Regarding intercultural competence, there are many terminologies: multiculturalism, cross-cultural adaptation, intercultural sensitivity, cultural intelligence, international communication, transcultural communication, cross-cultural awareness, intercultural communicative competence and cross-cultural competence [18,19]. According to Deardorff and Jones [20], there are some conceptualizations of intercultural competence that emphasize the interaction and the collaboration among individuals of different backgrounds in order to achieve a common goal without any kind of racist or discriminatory attitude [21]. Similarly, Esterhuizen and Kirkpatrick [22] describe intercultural competency as the cognitive, affective, and behavioral skills that support effective and appropriate interaction or a way of getting along with others. Finally, Deardorff [23] describes intercultural competence as an ongoing process. During this process, individuals should reflect and assess the development of their own intercultural competence over time, and should come to challenge the dominant knowledge and culture [24].

Currently, intercultural competence puts emphasis on individuals' acquisition of skills, attitudes, knowledge and ways of thinking that could assist them to smoothly move from ethnocentrism to ethnorelativism [25]. Hence, individuals who are highly interculturally competent have more ethnorelative cultural worldviews and cosmopolitan outlooks [26].

## 3. Methodology

The present study aims to investigate issues relevant to Greek teachers' global and intercultural competences through the following research questions: Which are teachers' perspectives on global and intercultural competences? Which practices do they use in order to induce diverse students to become globally and interculturally competent? What are teachers' training needs regarding global and intercultural competencies when teaching in multicultural settings?

The current research follows the principles of the constructivist worldview. Constructivism advocates that understanding of the world is obtained through experiences and interaction. Researchers who are guided by constructivism are interested in the complexity of views and they are based on participants' perspectives and experiences in order to export their results. The constructivist worldview has some implications that may affect the inquiry. The researcher should uncover and understand participants' beliefs and experiences about the studied issue, in order to form and develop his/her theory [27].

The sample of the study consists of thirteen teachers (N = 13) of primary school education and all of them were teachers of reception classes, Zones of Educational Priority (ZEP) at Attica Municipality. These classes are a 'special type' of classes where immigrant and refugee students attend. The participants were chosen through a convenient sampling process, as their viewpoints and beliefs were valuable for the conduction of the study [28]. Convenience or opportunity sampling is the most usual type of non-probability sampling in relative researches, and the main criterion for sample selection is the convenience to and resources of the researcher, based on the participants' having the characteristics needed for the research [29,30]. In this case, these characteristics were (a) teachers of reception classes; and (b) working in Greece. As Dörnyei and Csizér [29] note, convenience sampling is not entirely based on convenience but it is also somewhat purposeful in the sense that

participants need to strictly fit the criteria posed for the research. It is also worth noting that the study took place from March 2021 to July 2021.

### 3.1. Data Collection and Data Analysis

Semi-structured interviews were followed as the most appropriate option, since they allow the researcher to rephrase or clarify the questions he/she poses, they provide a detailed review of the studied issue and they reveal the personal experiences and the diverse opinions of the participants [31]. However, as a process they are time-consuming and the researcher should demonstrate patience during the preparation and the transcription of the interviews [32].

Regarding the questions of the present study, they were organized in four distinctive groups. The first one was about the demographic and the professional elements of participants. The second part was about their perspectives on global and intercultural competences, and the third one was about the practices they apply in order to motivate students to develop global and intercultural competences. The questions of the last part were about teachers' training needs on global and intercultural issues.

Finally, all semi-structured interviews took place remotely, either through Webex meetings or through Skype. Each of them lasted approximately twenty minutes.

After transcription, the data were divided into sections based on a coding system that emerged through the texts. Moreover, the researcher worked based on the deductive method, which is considered as a common method at qualitative researches [33].

Data analysis proceeded according to the following steps. The first step was the transfer of interviews, which is the procedure of transcription of the digital records to written texts. The second step was the familiarization with the data and the collection of elements of the interviews that respond to each question. The third step was coding, meaning that each part of the text was given a conceptual definition, in other words a code. The fourth step involved the transition from coding to sections. At this step, codes that were similar in meaning were merged and edited, resulting in more general concepts and subsections. The last step was the presentation of the data [33].

Finally, it is worth mentioning that the language used for the interviews was Greek, since it was the mother tongue of all the participants, and they found it easier to express their thoughts and perceptions in their first language.

### 3.2. Validity, Reliability, Ethical Considerations

The use of reliability and validity are crucial for qualitative research. Reliability refers to the consistency of the presentation of the findings, and validity refers to the accuracy and the sincerity of research findings [34]. Regarding the reliability of this study, two pilot interviews were conducted in order to spot any differences in the answers of educators. Also, the researcher checked if the answers of the participants were reliable according to the literature review. The two participants' answers gave similar data, so they were evaluated as valid and reliable. Additionally, the questions of the interview were based on researchers' personal concerns and on the educational, social and professional reality as seen from the literature review. In addition, the questions served the main objectives of the research attempt. Finally, the interviewer avoided interviews with educators she knew in person [35].

A significant feature of research design also involves ethical issues [36]. Researchers should follow some rules in order to protect all of the participants [37]. This research adopted the procedural and relational ethics as proposed by Tracy [38]. Procedural ethics encompasses the idea that the participants were informed of confidentiality, voluntary participation as well as the content and nature of the research, along with the ability to withdraw at any time beforehand. Moreover, informed consent was achieved by their voluntary participation, given the circumstances and the digital form of the interview, in advance by their deciding to fill in a form forwarded to them and after having received the explanatory message regarding the aforementioned issues. Due to the difficult circumstances, there was

not a signed consent form. Regarding relational ethics, mutual respect and dignity of both the researcher and the participants was assured. Both parties were mindful and respectful of each other's. Lastly, directing or influencing their answers was avoided. On the contrary, they were free to express themselves however they felt appropriate.

*3.3. Findings*

In this section the findings of the research will be presented by starting with the demographic data and then moving on with to the findings per each research question: teachers' perspectives on global and intercultural competences, practices applied to induce students to become global and intercultural competent and training needs.

Concerning their demographic characteristics, all the participants were female, and their age ranged from 25 to 32 years. Regarding their general educational experiences, they ranged from two to six years, whereas their experience in the Zones of Educational Priority (ZEP) ranged from one to two years. Six out of thirteen participants have a postgraduate degree. Moreover, five of them have educational experience in multicultural environments.

At the beginning, teachers were asked about what specific skills and competences students should develop in intercultural environments. Their answers included linguistic skills, communication and cooperation skills, adaptability, respect, tolerance, social skills, solidarity, mutual respect, empathy and understanding of the principles of equality and parity. Specifically:

P2: "They (students) should develop beneficial social skills such as solidarity, mutual respect . . . empathy towards other people and to comprehend the principle of equality . . . parity".

P4: "I think that it is beneficial for the students to develop their sociability, cooperation, adaptability and acceptance of the difference".

P1: "The linguistic skills is a really important part . . . to exist a code of communication in which they can understand each other. But the most important for them is to develop a culture of acceptance of the different . . . an attitude of mutual respect and mutual acceptance towards the different".

Participants were also asked about their knowledge regarding the terms of "global competence" and "intercultural competence". Twelve out of thirteen interviewees stated that they had never before heard about global competence. Moreover, three participants were familiar with the term of intercultural competence, without, though, being able to define it. Most of their answers were about interaction, understanding and respect for other cultures. Specifically:

P6: "The truth is that I haven't heard of them . . . however I believe it is to put yourself in the other person's shoes and have an open mind . . . ".

P8: "No . . . I assume . . . it has to do with the way we interact with other cultures?".

P2: "No . . . global or intercultural competence . . . maybe is the competence to be able to act in an environment in which there is heterogeneity, where different cultures coexist . . . and to be able to live and coexist in it in harmony".

Regarding the importance of developing these competences, all the participants agreed on their importance. Specifically:

P1: "(It is) very important. At first glance, I think children with refugee or immigrant backgrounds may need it more, but in fact I think native children need it too, because at some point everyone . . . may find themselves . . . in another country so it would be good to have these competences".

P8: "Yes, I think it is very important . . . it is of the same importance (for all children) . . . we all live in a multicultural environment, so it is important that we should all be able to interact appropriately".

Supplementary to the previous question, participants were asked to express their opinions about the development of students' competences through schooling. All the participants shared the aspect that school environment should play a crucial role in the

development of the competences. However, many of them expressed their concern about the extent to which the school can really contribute towards this direction. Specifically:

P8: "I believe that school can play a significant role. Because it constitutes this kind of environment. Multicultural".

P4: "I would characterize it as a great competence, which the school should develop . . . but I do not know to what degree it can happen".

P5: "I believe that attempts are being made, in order all these skills to be developed through school, of course we are at an early stage. We still have work to do regarding this issue".

The next sector of the interview was relevant to teaching practices that participants use, in order the guide students to develop specific parts of global and intercultural competencies. The first question sought to reveal which practices and methods participants use in order to motivate their students to explore their world, both locally and globally. Participants' practices were the implementation of projects that approach the issue of multiculturalism, discussion and dialogue, use of appropriate texts, videos and educational material, role playing, collaboration with students from other regions, cultures and countries. Specifically:

P1: " . . . by giving (the teacher) many opportunities through the coexistence in a class, the coexistence of different cultures I mean...".

P4: "Through various programs, through projects, through collaborations maybe with other schools . . . ".

P8: "Recently we . . . worked on how children live in other countries . . . this was done through video . . . ".

P5: "Through the projects that I use . . . ".

P3: "Discussing and telling everyone their own experiences and habits . . . ".

P8: "I will tell you again games and especially role play...".

The last part of the interview sought to investigate teachers' needs on global and intercultural competencies. According to their answers, they are not prepared to teach these issues. Specifically:

P2: "I think that they are not very prepared, not much training has been done on these issues, that is, we find ourselves many times in front of issues that we do not know how to handle, how to approach them. More training is needed on these topics".

Moreover, all participants evaluated that relevant training programs would be really beneficial for teachers' improvement. Furthermore, participants proposed training programs, the topics of which should be relevant to the refugee crisis in Greece and its consequences, teaching practices and methods about global and intercultural topics, intercultural education, religious issues. Specifically:

P2: "I think it is very important (training) for the refugees, because... in Greece it is the burning issue".

P5: "Yes, it should be (training) . . . I would like to include more didactic and practical issues, how we will help the children through exercises, activities, projects to achieve all that we said above".

Afterwards, participants were asked to share their perspectives on the most appropriate training provider entity. Most of them proposed the Greek Ministry of Education, the European institutions and schools or even educators that have relevant knowledge and experience. Specifically:

P6: "Certainly someone European institution, like the Council of Europe for example".

P5: "A university or even a colleague who is well educated".

P3: "I think the Ministry of Education, but maybe in collaboration with a university . . . ".

Additionally, participants were called to make a suggestion about the duration of these training programs. Their answers vary, from one month to one year or even to never stoping. Simultaneously, there were mentions of the importance of the content of the training, rather than the duration. Specifically:

P6: "I think it should not be just a six-month, two-month, three-month training that ends. They should have a smooth sequel . . . ".

## 4. Discussion

The research revealed some interesting findings. To begin with, the participants state that they are not familiar with the term of global competence, but they seem to be more familiar with the term of intercultural competence, since they recognize some basic goals and principles of the intercultural competence, such as mutual respect, understanding and effective interaction among individuals of different cultures. However, despite the fact that they cannot define global competence, they do have awareness of some major dimensions of it. Taking the OECD definition into consideration, there are some more dimensions of it which were not mentioned, such as inequality, environmental justice, poverty, trade, constructive action, sustainability and well-being [4,13,19]. Researchers and educators seem to have recently started to be concerned and to closely study students' global competence. According to the OECD [4], priority was given to global citizenship and global competencies in 2018. This factor may explain the limited knowledge of teachers about this term. On the contrary, intercultural competence is a more familiar term for the participants, as it has been used by researchers since 2003 or even earlier [39–41]. Similarly, according to Simpson and Dervin [42] (p. 672), "*the notion of interculturality has been with us since the beginning of time and was found under different guises since classical antiquity*". In addition, all the international organizations such as the OECD, the UNESCO, the Council of Europe have developed policies towards this direction and have imposed ways of discussing and even assessing interculturalism to the whole world [42].

Moreover, all participants evaluated the principles of global and intercultural competences as of crucial importance for citizens in multilingual and multicultural environments. They also recognized the significant role the school plays in the development of these competences. Furthermore, they comprehend the importance of global and intercultural competencies for the thriving of modern societies. Researches also recognize the importance of global and intercultural competencies at the evolution and prospering of multicultural societies [13,16]. Additionally, studies evaluate the development of global and intercultural competencies by students as one of the most significant educational outcomes [4,13,16]. However, the participants state that education does not seem to have sufficient results at students' developing of these competences, as participants' answers seem that students do not acquire global and intercultural competences.

As far as their teaching practices, most teachers try to apply a range of methods or practices for teaching students how to develop global and intercultural competences. Discussions, appropriate texts, stories, videos, teaching through gaming and projects are some of the most popular practices. These methods and practices are in accordance with other similar studies [39]. Team games and project-based learning are among the recommendations of the OECD [4] for an effective teaching of global and intercultural competences, practices that are practiced by some participants, also. Moreover, bibliography suggests a range of ways and practices, in order to teach students global and intercultural competences efficiently. These suggestions [4,39] refer to students' active engagement in local and global issues and in cultural subjects. It is then important to share one's understanding of inequalities and injustices through constructive dialogue and discussion [43]. Particularly, instructional approaches include structured discussion, organized discussion, and current events discussion, which help students delve deeply into a topic, change their minds, find out what is happening around the world and in their local environment as well [39]. These practices, especially the discussion one, are widely used by the participants in order to raise awareness of pupils on intercultural issues. It is also worth mentioning that global education requires several approaches to learning and particularly: hope, memory and dialogue. Hope represents the need to move forward in one's life, but not without remembering and knowing past injustices (memory) [44].

The study also examines potential teachers' needs regarding their readiness to educate students appropriately, in order to develop global and intercultural competences. All participants agree that educators require extra training in order to acquire knowledge and experience in effective relevant teaching practices and methods. Teachers' knowledge and readiness to teach global and intercultural competencies affect the extent that students develop these competencies [9]. Moreover, educators' efficiency at teaching global and intercultural abilities depends on their comprehension and development of these competencies, as well as their awareness on different cultural characteristics and ways to handle them effectively in a multicultural classroom [24,45–47].

Furthermore, the participants agree with Dunn [47] regarding the importance of teachers' guidance by experts in this field. Teachers' appropriate and adequate knowledge and skills are also issues that concern researchers, as they may widely influence the way that teachers guide students to global and intercultural awareness [47–49].

Regarding the training programs, there are some basic elements that should be taken into consideration while designing an appropriate training program: finding a common understanding (trainers and their organizations should discuss and work together effectively on all issues regarding training), learning-needs analysis (in order to meet the participants' training needs and cover the gaps in their learning), developing a contextualized competency model (learning goals that identify knowledge, capacities and competencies), empowering adult learners (individual support, previous experiences), and devising a training structure (in coherence with principles and goals) [49,50].

In conclusion, the national and international interest in intercultural and global issues is on the increase. Students, teachers, parents, citizens and educators will be called on to participate in relative programs and modules, change attitudes, and initiate the appropriate practices. However, according to Simpson and Dervin [42], we should critically approach issues regarding the ideologies, the real purpose, the silenced and not silenced voices, the social, political, and economic aspects that are promoted through these interventions. In practice it is observed that intercultural and global competencies are not promoted by all the modules and subjects, and teacher training usually responds to particular commitments of certain areas and does not have a practical application [51]. In addition, the material used is mostly Western-centric [42], which means that we need to go beyond "abyssal thinking" in order to recognize that not all knowledge is based on Western epistemologies and there is still knowledge that is absent in schools and academia [52].

## 5. Research Limitations and Future Proposal

The conduction of a study involves a demanding and time-consuming procedure. Its organization should be well-prepared and the researcher should be ready to overcome any possible unexpected obstacles and barriers. However, as the literature proves, it is inevitable for almost every study to come up against limitations and shortcomings, no matter how organized or well-prepared it may be [30]. This particular research is no exception. The main limitation of the research is that the results are not generalizable for the whole population.

In addition, the research took place from March 2021 to July 2021 and the interviews were conducted through Webex and Skype applications due to the COVID-19 pandemic. Thus, no verbal signs (such as gestures or body movement) could have been taken into consideration in an appropriate way.

Another factor that derives from distance interviewing and that may affect the results of the study was the use of applications at the conducting of the interviews. Despite the fact that participants' anonymity was guaranteed by the researcher, there were participants who expressed their concern regarding their possible identification, due to the use of the Webex application. The solution to this concern was either the use of Webex through an internet page where someone could participate in the meeting without his/her personal information being recorded or the use of other applications that were not connected to the Greek School Network.



One more research limitation is the result of research design and the research tool itself, since semi-structured interviews both allow participants to express their subjective opinion and the researcher to interpret the data according to her own experiences and perceptions.

Lastly, factors that may limit the generalization of the present findings are the delimited time and place where the research took place. The inquiry was carried out in the region of Attica in a specific time period, so the study is formed by these two factors.

Regarding future exploration, this study could constitute a starting point for further relevant studies that may examine the issue more deeply. For instance, teaching global competence with a social justice mission and guiding students to comprehend critical and postcolonial approaches to its practice presents a major research challenge as well. According to Sorrells [44], critical intercultural and global communication participants should focus as well on the unequal forces of globalization in terms of technology (the fourth industrial revolution), financial power, and demographic and climate changes in the light of colonialism and hegemony. This differs from traditional intercultural courses that focus primarily on skills-based, interpersonal interactions in that the focus moves from micro to macro [44].

In addition, the research took place in Attica Municipality, so a sample of educators who teach at other geographical regions and in general classes (and not only at reception classes) both in primary and secondary education might be helpful in order to depict more general aspects and perceptions. Moreover, Attica is a multicultural and multilingual prefecture.

Finally, the outcomes of this study could be used as a tool for reflection and guidance for teachers and teacher educators who intend to develop not only their global teaching competencies but also students' competencies [53].

**Author Contributions:** Conceptualization, G.K. and Z.K.; methodology, G.K.; software, G.K.; validation, Z.K., G.K. and N.P.; formal analysis, G.K.; investigation, G.K.; resources, G.K. and Z.K.; data processing, G.K. and Z.K.; writing—original draft preparation, Z.K. and G.K.; writing—review and editing, Z.K., G.K. and N.P.; visualization, G.K.; supervision, Z.K. and N.P. All authors have read and agreed to the published version of the manuscript.

**Funding:** This research received no external funding. Only the paper received financial support from HOU for its publication.

**Institutional Review Board Statement:** The study was conducted according to the guidelines of the Declaration of Helsinki, and approved by the Institutional.

**Informed Consent Statement:** Informed consent was obtained from all subjects involved in the study.

**Data Availability Statement:** The submission of the thesis has been accepted and filed at DSpace and has the following identification: https://apothesis.eap.gr/handle/repo/51486 (accessed on 29 January 2022).

**Acknowledgments:** The authors of the article would like to thank the teachers who participated in the study; these teachers were inspired by two projects: (a) the training sessions of the Global Competences for Teachers Project (G.C.T.E., 2019–2022) which is an Erasmus + project, (b) "Migrant Children and Communities in a Transforming Europe" (MICREATE acronym) project which is an Horizon project (2019–2022).

**Conflicts of Interest:** The authors declare no conflict of interest.

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
