# Peer review of "Teachers’ Global Perceptions and Views, Practices and Needs in Multicultural Settings"

_education, doi:10.3390/educsci12040280_

Round 1

Reviewer 1 Report

It was a pleasure to read the article submitted for review. The text is interesting and important in the area of teachers' functioning in a diversified world and reference to their global and intercultural competences. I have no comments on the theoretical part, but if there was space, it would be worth adding some more recent references to similar research. I can't quite agree with the content of the 'methodology' section. It seems at times to be imprecise in presenting key aspects concerning the presented study. The method, procedure, time of research and ethical issues should be improved (they are described, but it is not clear from the description whether the participants actually gave their consent and in what form). In Findings, it would be useful to indicate at the beginning the thematic areas that will be presented in the following sections. In the last paragraph of the article, it is advisable to justify the sentence: "However, in Greece teachers seem not to have acquired global or intercultural com-352 petences." This is a very strong statement which does not reflect all teachers in Greece but only the results of the sample of teachers who participated in the survey. Best regards

Author Response

Dear reviewer, 

I would like to thank you for the comments to the point, which have been taken into consideration in order to improve our text. All the changes are highlighted in red. 

Kindly

Reviewer 2 Report

This is a very relevant subject matter in the contemporary globalised environment where dominant narratives are being maintained as the 'norm'. The critical exploratory dialogue needed to critique such hegemonic stances informing social constructions is all too under represented. This research shows the analysis of the understanding and meanings of such terminologies through the powerful use of linguistics and language, can be mis-understood, mis/interpreted and unknown.  This goes to the 'heart' of the knowledge development and acquisition - the teachers.

Future exploration - in relation to the teachers pedagogy methods and practice mentioned, possibly an exploration of 'critical pedagogy' may be an area of interest?

Just a few minor changes:

  • A proof read to check the odd spelling; grammatical presenting of phrases - sometimes colloquial terms were used (e.g. pretty certain..).
  • Sometimes the sentence structure and paragraphs used more words than needed, try a 'less is more' approach to present the discussion points.
  • There were parts that were explicitly explained that are not necessarily needed. e.g. in sections 3.1 and 3.2 you explain to the reader what certain functions are.

Otherwise a very interesting read. 

Author Response

Dear reviewer,

I would like to thank you for the comments to the point, which have been taken into consideration in order to improve our text. All the changes are highlighted in red. Except for the words and sentences in red, the whole text has been revised and checked for any grammar and syntax mistakes. 

Kindly

Kindly

Reviewer 3 Report

From the overall presentation I would say that interesting research work has been done. The topic is also important for the readers of the journal. However, I have a few more significant challenges with the paper. 

The theoretical part remains at a modest level. At this stage, it does not yet provide an in-depth review of the previous literature. It is more a description than analysis. Therefore, a more detailed explanation of theoretical background and research design needs to be supplemented for this paper to be published. You should include some hypotheses and test them. 

“3. Methodology” section should explain in more detail how the data were collected and a broader description of the sample of participants should be made. This section is lacking information on the participant recruitment method, namely: a) the recruitment date range (month and year), b) a description of any inclusion/exclusion criteria that were applied to participant recruitment, c) a statement as to whether your sample can be considered representative of a larger population. 

I see the sampling to be the most limiting factor of the presented paper. 

It would be appropriate to specify in more detail how this research differs from the already published paper that deals with a similar topic. To increase the significance of the results, the discussion part should embrace the differences and similarities among your findings and those of other scholars. 

You need to improve the practical and academic implications. 

However, the paper has to underline the limits of the research and future work. 

Author Response

Dear reviewer,

I would like to thank you for the comments to the point, which have been taken into consideration in order to improve our text. All the changes are highlighted in red. However, I would like to express our opposition to the use of hypotheses, given that we do not usually use them while conducting qualitative researches. 

Kindly

Round 2

Reviewer 3 Report

Dear Authors, 

In the revised version, the manuscript has been extended and improved. 

Best regards 

Author Response

Dear reviewer,

thank you for your valuable comments on our work.

I would like to say that we have taken the comments of all the reviewers into account, even the additional comments of the second round.  To begin with, there have been changes and supplementary information regarding the method, procedure, time of research, language of interviews, sampling process, ethics, limitations, future proposal and existing situation in Greece (policies and flows). Regarding the presentation of findings, there has been added a short introduction part, according to the comments of the 1st reviewer, and there have been some corrections regarding the use of "overgeneralized" aspects.

What is more, critical pedagogy aspects have been added as well, according to your comments, in different parts (e.g Finally, teachers are members of the dominant culture, so they may face difficulties to apprehend the challenges, fears and needs of individuals who are called to get adjusted to culturally diverse environments [11], During this process, individuals should reflect and assess the development of their own intercultural competence over time, and should come to challenge the dominant knowledge and culture [24], the national and international interest in intercultural and global issues is on increase. Students, teachers, parents, citizens and educators will be called to participate in relative programs and modules, change attitudes, and implicate appropriate practices. However, according to Simpson and Dervin [42], we should critically approach issues regarding the ideologies, the real purpose, the silenced and not silenced voices, the social, political, and economic aspects that are promoted through these interventions. In practice it is observed that intercultural and global competences are not promoted by all the modules and subjects and teacher training usually responds to particular commitments of certain areas and does not have a practical application [51]. In addition, the material used is mostly Western centric [42], which means that we need to go beyond “abyssal thinking” in order to recognize that not all knowledge is based on Western epistemologies and there is still knowledge that is absent in schools and academia [52].

It is also worth mentioning that we focused on the OECD's definition of Global Competence, given that this was chosen by the leader of the Erasmus Plus Project, in which we participated. 

Regarding the presentation of the findings, it is a common practice for us, as researchers, to present participants' aspects without making any comments, according to the research questions, and then at the discussion part we follow the comparison approach (similarities, differences, other relative aspects etc) according to relative bibliography and similar researches.

Finally, the discussion part has been enriched by extra bibliography and we tried to use the proposed material by you as well, which was really significant and helpful in order to give our topic a deeper insight and a more critical approach.

Thank you for your time and concern!